# Distinct patterns of brain Fos expression in Carioca High- and Low-conditioned Freezing Rats

Laura A. León[1,2,3], Marcus L. Brandão[1], Fernando P. Cardenas[4], Diana Parra[4], Thomas E. Krahe[2], Antonio Pedro Mello Cruz[5], J. Landeira-Fernandez[2]*

**1** Laboratory of Neuropsychopharmacology, FFCLRP, Behavioral Neuroscience Institute (INeC), São Paulo University, Campus USP, Ribeirão Preto, São Paulo, Brazil, **2** Department of Psychology, Pontifical Catholic University of Rio de Janeiro, Rio de Janeiro, Brazil, **3** Programa de Psicología, Universidad Sergio Arboleda, Bogotá, Colombia, **4** Laboratorio de Neurociencia y Comportamiento, Universidad de los Andes, Bogotá, Colombia, **5** Institute of Psychology, University of Brasilia, Brasilia, Brazil

* landeira@puc-rio.br

## Abstract

### Background

The bidirectional selection of high and low anxiety-like behavior is a valuable tool for understanding the neurocircuits that are responsible for anxiety disorders. Our group developed two breeding lines of rats, known as Carioca High- and Low-conditioned Freezing (CHF and CLF), based on defensive freezing in the contextual fear conditioning paradigm. A random selected line was employed as a control (CTL) comparison group for both CHF and CLF lines of animals. The present study performed Fos immunochemistry to investigate changes in neural activity in different brain structures among CHF and CLF rats when they were exposed to contextual cues that were previously associated with footshock.

### Results

The study indicated that CHF rats expressed high Fos expression in the locus coeruleus, periventricular nucleus of the hypothalamus (PVN), and lateral portion of the septal area and low Fos expression in the medial portion of the septal area, dentate gyrus, and prelimbic cortex (PL) compared to CTL animals. CLF rats exhibited a decrease in Fos expression in the PVN, PL, and basolateral nucleus of the amygdala and increase in the cingulate and perirhinal cortices compared to CTL animals.

### Conclusions

Both CHF and CLF rats displayed Fos expression changes key regions of the anxiety brain circuitry. The two bidirectional lines exhibit different pattern of neural activation and inhibition with opposing influences on the PVN, the main structure involved in regulating the hypothalamic–pituitary–adrenal neuroendocrine responses observed in anxiety disorders.

**Data Availability Statement:** All raw data are available from the Dryad database. DOI: 10.5061/dryad.h70rxwdg7 Reference: León, Laura et al. (2020), Distinct patterns of brain Fos expression in

Carioca High- and Low-conditioned Freezing Rats, Dryad, Dataset, https://doi.org/10.5061/dryad.h70rxwdg7.

**Funding:** This work was supported by grants from the National Council for Scientific and Technological Development (CNPq) and Rio de Janeiro State Research Foundation (FAPERJ) to JLF. This study was also financed in part by the Coordenação de Aperfeiçoamento de Pessoal de Nível Superior - Brasil (CAPES) - Finance Code 001. LAL had a doctoral fellowship from CAPES.

**Competing interests:** The authors have declared that no competing interests exist.

# Background

The concept of anxiety encompasses a series of defensive behaviors and neurophysiological responses that individuals present when faced with a potentially threatening situation. These responses are mediated by a set of neurocircuitries that have been shaped by natural selection because of their adaptive function in protecting the individual from danger [1]. However, these responses can represent a pathological condition when they occur in excess or disproportionally to the threatening situation without any apparent adaptive function [2]. Anxiety disorders are among the most prevalent mental disorders [3] and are mediated by neurocircuitries that underlie adaptive behavioral and neurophysiological responses [4, 5].

Several rodent models have been developed to investigate the possible etiological mechanisms that underlie anxiety disorders [6–9]. Among these, fear conditioning in response to contextual cues is a well-studied experimental model of the aversive expectations of danger that are observed in patients suffering from generalized anxiety disorder [10, 11]. Furthermore, considerable evidence indicates that contextual fear conditioning in rats involves neural circuitries similar to that associated with anxiety disorders in humans [12–14]. In this model, rodents receive brief footshocks (unconditioned stimuli) minutes after being placed in a novel chamber, and, when returned to the same chamber 24 h later, they present a typical freezing response to contextual cues associated with the footshocks [15, 16]. Employing a bidirectional selective breeding procedure, our group developed two lines of animals with high and low conditioned freezing responses in the contextual fear conditioning paradigm [17]. These lines are termed Carioca High- and Low-conditioned Freezing rats, respectively (CHF and CLF).

Over the last decade, several works have established the Carioca lines as an animal model for the study of anxiety related disorders. For instance, CHF rats display more anxious like behaviors than CLF rats in the elevated plus maze, less social interactions than normal animals, and higher plasma corticosterone concentrations compared to both CLF and CTL animals [18–20]. On the other hand, no behavioral differences were found between CHF and CLF animals in the forced swim test [18], suggesting a dissociation between anxiety and depression traits in the Carioca lines. Moreover, cognitive and memory performance of CHF and CLF rats both in the object recognition task and the Morris water maze test were similar to normal animals [18, 21].

The freezing response to contextual cues previously associated with footshocks observed in the Carioca lines has also been pharmacologically validated as an adequate model of anxiety disorders. We have recently demonstrated that classic anxiolytic benzodiazepines, such as midazolam reduces the amount of conditioned contextual freezing responses of CHF rats [22]. Similar results were observed with non-benzodiazepine anxiolytics, microinjections of ketaserin, a 5-HT2A/2C receptor antagonist, in the infralimbic mPFC cortex reduced the amount of freezing responses of CHF rats in the contextual fear conditioning paradigm [23]. Moreover, both systemic and infralimbic cortical injections of ketanserin increased the number of open arm entries and time spent in the open arms of CHF animals in the elevated plus maze [23].

According to these behavioral and pharmacological findings, it is likely that CHF and CLF rats display different neural activity patterns in response to contextual cues previously associated with footshocks. One way to explore this possibility is to evaluate changes in Fos expression in specific brain regions. Fos is an immediate early gene product that is synthesized in neurons through an increase in second messengers, such as cyclic adenosine monophosphate and calcium ions. Fos immunochemistry is a widely and well-established method employed to mark neuronal activity with high spatial resolution when rats are exposed to aversive stimuli [24, 25]. Thus, the present study performed Fos immunochemistry to investigate whether the

CHF and CLF breeding lines exhibit different levels of neuronal activity in specific brain areas compared to control animals (CTL). Fos protein immunoreactivity was assessed in serial sections of different brain structures in CHF, CLF, and CTL animals after exposure to contextual stimuli that were previously associated with footshock. Here we hypothesized that CHF and CLF animals would display different Fos expression profiles in key regions of the anxiety brain circuitry when exposed to the aversive contextual cues.

## Materials and methods

### Animals

All animals were bred in the animal facilities of the Psychology Department, Pontifical Catholic University (PUC-Rio), Rio de Janeiro, Brazil. We used male adult rats selectively bred for high (CHF, n = 10) and low (CLF, n = 10) contextual fear conditioning according to previously described procedures [17]. Non-selectively bred Wistar rats were used as a control group (CTL, n = 12). All animals were born and maintained in the colony room of the PUC-Rio Psychology Department under controlled room temperature (24˚C ± 1˚C) and a 12 h/12 h light/dark cycle (lights on 7:00 AM–7:00 PM). The rats were housed in groups of five to seven per polycarbonate cage (18 cm × 31 cm × 38 cm) according to their respective lines with food and water available *ad libitum*. The experiment was conducted during the light phase of the light/dark cycle. The rats were tested at 3–4 months of age and weighed 190-330g. The experimental procedures were performed in accordance with the guidelines for experimental animal research that were established by the Brazilian Society of Neuroscience and Behavior (SBNeC) and National Institutes of Health *Guide for the Care and Use of Laboratory Animals*. Animal handling and the methods of sacrifice were reviewed and approved by the Committee for Animal Care and Use of the Pontifical Catholic University of Rio de Janeiro (PUC-Rio) protocol no. 20/2009.

### Apparatus

The contextual fear conditioning protocol was conducted in four observation chambers (25 cm × 20 cm × 20 cm), each placed inside a sound-attenuating box. A red-light bulb (25 W) was placed inside the box, and a video camera was mounted on the back of the observation chamber to observe the animal's behavior on a monitor that was placed outside the experimental room. A ventilation fan that was attached to the box supplied 78 dB background noise (A scale). The floor of the observation chamber consisted of 15 stainless-steel rods (4 mm diameter) that were spaced 1.5 cm apart (center-to-center), which were wired to a shock generator and scrambler (Insight, São Paulo, Brazil). An interface with eight channels (Insight) connected the shock generator to a computer, which allowed the experimenter to apply an electric footshock. Ammonium hydroxide solution (5%) was used to clean the chamber before and after each subject.

### Procedure

The contextual fear conditioning procedure consisted of an acquisition and a test session. During acquisition, each animal was placed in the observation chamber for 8 min. At the end of this period, three unsignaled 0.5 mA, 1 s electric footshocks were delivered with an intershock interval of 20 s. Three minutes after the last footshock, the animal was returned to its home cage. The contextual fear conditioning test session was conducted approximately 24 h after training. This test consisted of placing the animal for 8 min in the same chamber where the

three footshocks were delivered the previous day. No footshock or other stimulation occurred during this period.

## Fos protein immunochemistry

Two hours after the conditioned fear test (i.e., the interval that is required for the synthesis and accumulation of Fos protein; [26]), the animals were deeply anesthetized with an overdose of urethane (1.25 g/kg, intraperitoneal; Sigma-Aldrich, St. Louis, MO, USA) and intracardially perfused with 0.1 M phosphate-buffered saline (PBS) followed by 4% paraformaldehyde in 0.1 M PBS (pH 7.4). The brains were removed and stored in 30% sucrose in 0.1 M PBS for cryo-protection. The brains were then frozen in isopentane (-40˚C) and sliced in a cryostat (-19˚C). As described in our previous studies [27–29], coronal 40 μm cryostat sections were collected in 0.1 M PBS and subsequently processed free-floating according to the avidin–biotin system using the Vectastain ABC Elite peroxidase rabbit IgG kit (Vector, USA). All reactions were carried out under agitation at 23±1˚C. The sections were first incubated with 1% $H_2O_2$ for 10 min, washed four times with 0.1 M PBS (5 min each), and incubated overnight with rabbit polyclonal primary IgG against Fos (Santa Cruz Biotechnology, Santa Cruz, CA, USA). The next day, the sections underwent a series of three 5-min washes and were then incubated for 1 h with secondary biotinylated anti-rabbit IgG (H+L; Vectastain, Vector Laboratories). After another series of three washes in 0.1 M PBS (A and B solution, ABC kit, Vectastain, Vector Laboratories), the sections were incubated for 1 h with the avidin-biotin-peroxidase complex in 0.1 M PBS and then washed again three times in 0.1 M PBS. Fos immunoreactivity was revealed by the addition of the chromogen 3,3'-diaminobenzidine (DAB; 0.02%, Sigma, St. Louis, MO, USA) to which 0.04% hydrogen peroxide was added before use. Finally, the sections were washed twice with 0.1 M PBS.

Tissue sections were mounted on gelatin-coated slides and dehydrated, and Fos-positive neurons were counted by bright-field microscopy (Olympus, BX-50, 100× magnification, coupled to a Leica DFC320 video camera. The anatomical localization of Fos-positive cells was determined based on the Paxinos and Watson [30] stereotaxic rat brain atlas. The images were scanned and analyzed using ImagePro Plus 6.2 software (Media Cybernetics, Bethesda, MD, USA). The system was calibrated to ignore background staining. All brain regions were counted bilaterally (7–12 animals per region), and the mean was calculated for each structure. Fig 1 shows the anterior-posterior section coordinates of the studied brain structures.

## Statistical analysis

The data are presented as mean ± standard error of the mean (SEM). A one-way analysis of variance (ANOVA) was employed to analyze the percent of time freezing among the CTL, CHF, and CLF animals during test session. In order to analyze the Fos-positive cells results among the three animal groups, a one-way ANOVA was also conducted separately for each defined brain region. When statistical significance ($p < 0.05$) was obtained with an ANOVA, the Fisher's Least Significant Difference (LSD) post-hoc test was used to assess specific group differences.

## Results

Fig 2 shows the mean ± SEM percentage of time spent freezing among CHF, CLF, and CTL animals during the 8 min contextual fear conditioning test session 24 h after training. The one-way ANOVA indicated a significant difference among the three groups in conditioned freezing in response to contextual cues that were previously associated with footshock ($F_{2,35} = 37.53$, $p < 0.001$). The CHF line exhibited the highest conditioned freezing, and the CLF line

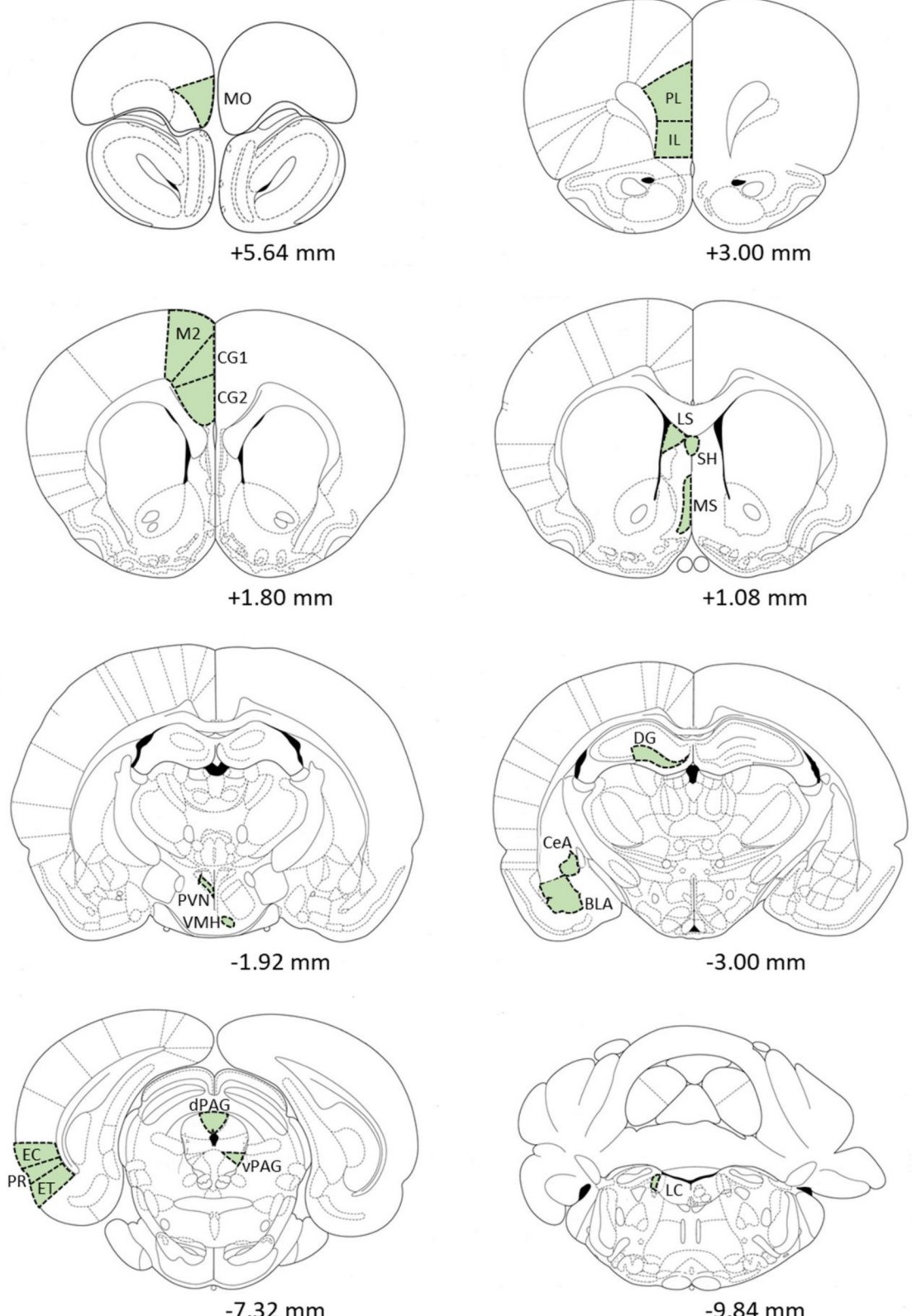

**Fig 1. Schematic diagrams, adapted from the atlas of Paxinos and Watson [30], showing the 20 areas (highlighted in green) where Fos expression was quantified.** Medial orbital cortex (MO), prelimbic cortex (PL), infralimbic cortex (IL), anterior cingulate cortex—subregion

1 (CG1), anterior cingulate cortex—subregion 2 (CG2), secondary motor area (M2), lateral septal nucleus (LS), septohippocampal nucleus (SH), medial septal nucleus (MS), paraventricular hypothalamic nucleus (PVN), ventromedial nucleus of the hypothalamus (VMH), dentate gyrus (DG), central amygdaloid nucleus (CeA), basolateral amygdaloid nucleus (BLA), ectorhinal cortex (EC), perirhinal cortex (PR), entorhinal cortex (ET), dorsal periaquedutal gray matter (PAGd), ventral periaquedutal gray matter (PAGv), and locus coeruleus (LC). Values under each diagram are bregma references.

exhibited the lowest conditioned freezing. The CTL line exhibited an intermediate level of freezing. This interpretation was confirmed by pairwise *post hoc* comparisons. CHF rats froze more than CTL and CLF animals, and CLF rats froze less than CHF and CTL animals ($p < 0.001$ for all comparisons).

Fig 3A and 3B shows the Fos expression smong the CTL, CHF, and CLF animals in different brain structures after exposure to contextual cues that were previously associated with

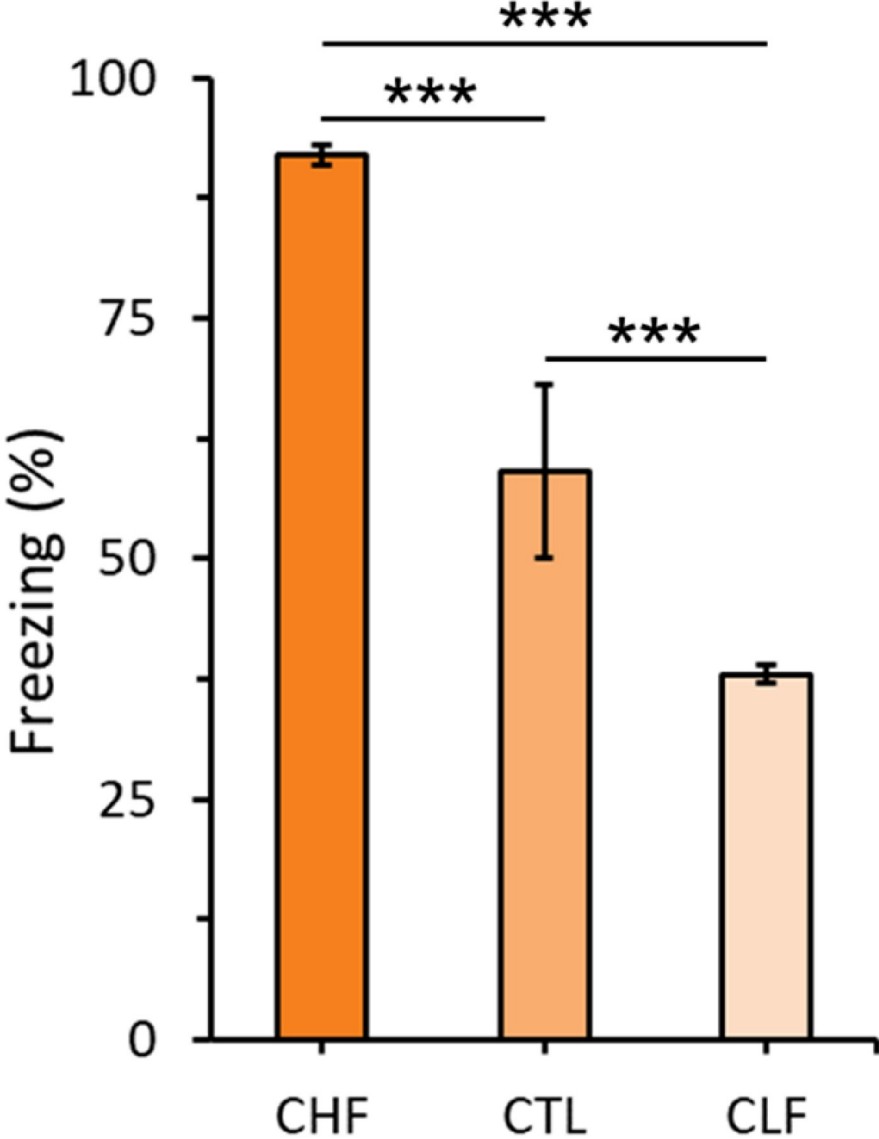

**Fig 2. Mean ± SEM percentage of time spent freezing in Carioca high freezing (CHF, n = 10), Carioca low freezing (CLF, n = 10), and control (CTL, n = 12) rats.** Graph depicts freezing responses 24 h after exposure to contextual cues that were previously associated with footshocks. Fisher's LSD post-hoc tests, $^*p < 0.001$ for all comparisons.

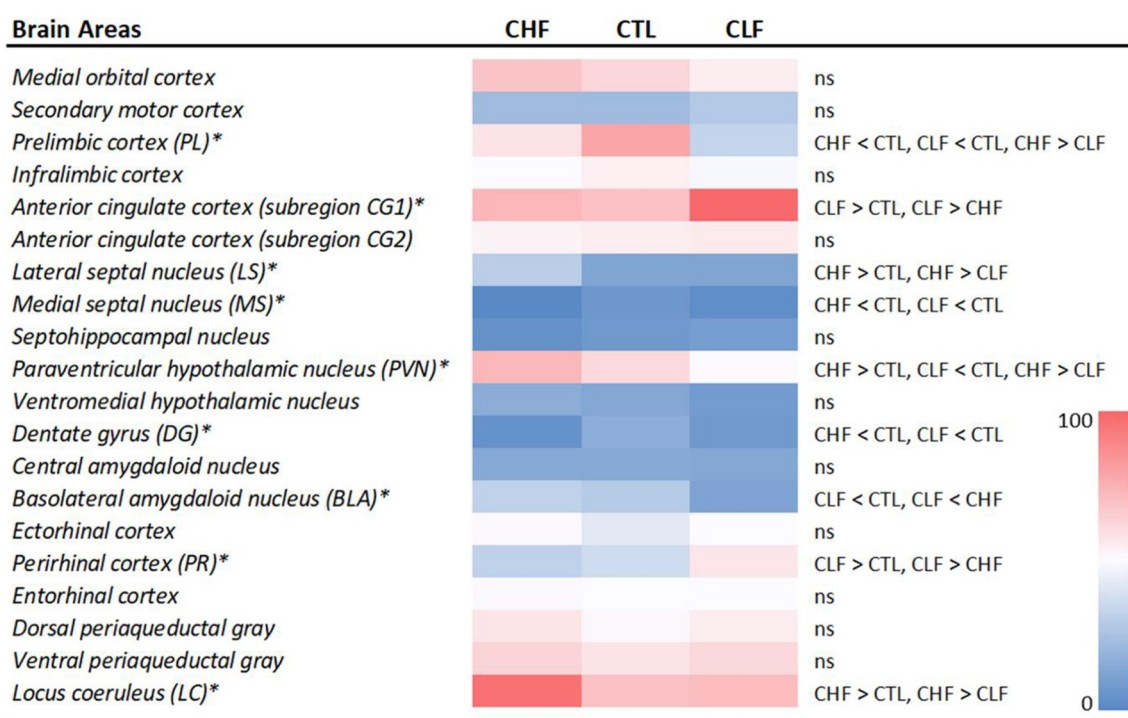

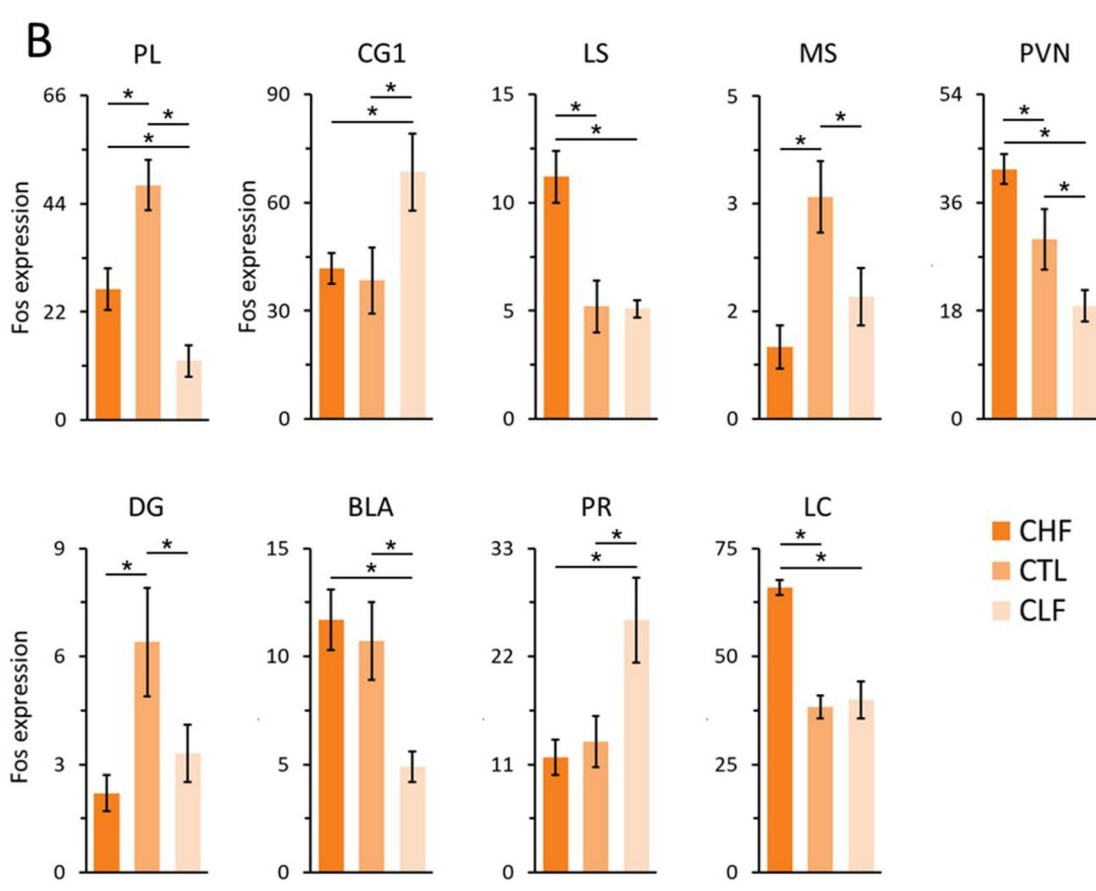

**Fig 3.** (A-B) Fos expression in different brain structures of Carioca high freezing (CHF), Carioca low freezing (CLF), and control (CTL) animals 2 h after exposure to contextual cues that were previously associated with footshocks (7–12 animals per region per group). (A) Mean Fos expressions (positive neurons/0.1 mm$^2$) between groups for each brain structure are presented as a heatmap. Changes in color from blue to red on the pseudo-color scale represent mean values of Fos-positive neurons/0.1 mm$^2$ ranging from 0 to 100 (please see S1 Table for mean and ± SEM values). The < and > signs indicate whether Fos expression in CHF, CLF, and CTL animals for a particular brain region was significantly smaller or greater compared to each other, and ns indicates the absence of statistical differences. (B) Bar graphs showing the mean (± SEM) number of Fos-positive neurons/0.1 mm$^2$ in CHF, CLF and CTL groups for brain regions that displayed significant differences in Fos expression depicted in A (structures marked with an asterisk). Fisher's LSD post-hoc tests, *$p < 0.05$ for all comparisons.

footshock. Statistically significant differences among groups were observed in the prelimbic cortex (PL; $F_{2,22} = 17.92$, $p < 0.001$). Pairwise comparisons revealed a decrease in PL Fos expression in both CHF and CLF rats compared to CTL rats, although Fos expression was higher in CHF animals than in CLF animals (all $p < 0.05$). The one-way ANOVA also revealed a significant difference in subregion 1 of the anterior cingulate cortex (CG1; $F_{2,27} = 3.42$, $p < 0.05$). CLF animals exhibited an increase in Fos activity in the CG1 compared to CTL animals ($p < 0.05$). The lateral septum (LS) and medial septum (MS) presented statistically significant differences in Fos expression (LS: $F_{2,22} = 14.26$, $p < 0.001$; MS: $F_{2,23} = 6.93$, $p < 0.001$). While pairwise comparisons revealed an increase in Fos expression in the LS only for CHF animals compared to controls ($p < 0.05$), a decrease in Fos expression in the MS as observed in both CHF and CLF rats compared to CTL (all $p < 0.05$). The one-way ANOVA indicated that the paraventricular nucleus of the hypothalamus (PVN) also presented a significant difference in Fos expression ($F_{2,22} = 9.51$, $p < 0.001$). Pairwise comparisons indicated an increase in Fos expression in the PVN in CHF rats compared to CTL rats and a decrease in Fos expression in CLF rats compared to CTL animals (all $p < 0.05$). The number of Fos-expressing cells in the dentate gyrus (DG) was also significantly different among groups ($F_{2,23} = 3.88$, $p < 0.05$). *Post hoc* analyses indicated that Fos expression in CHF and CLF animals were lower compared to CTL animals (all $p < 0.05$). Fos cell counts in the basolateral amygdala (BLA) were also significantly different among groups ($F_{2,27} = 8.47$, $p < 0.001$). When cell counts were compared between groups, CLF animals exhibited significantly lower Fos expression compared to CTL animals ($p < 0.05$). The one-way ANOVA of Fos expression in the perirhinal cortex (PR) indicated significant differences among groups ($F_{2,26} = 6.44$, $p < 0.01$). *Post hoc* comparisons revealed an increase in Fos expression in CLF animals compared to CLT animals ($p < 0.05$). Finally, an overall difference in Fos expression was observed in the locus coeruleus (LC) among groups ($F_{2,27} = 3.56$, $p < 0.05$). Pairwise comparisons indicated an increase in LC Fos expression in CHF rats compared to CTL rats ($p < 0.05$). No other significant differences were found. Examples illustrating the observed differences in Fos expression between groups are depicted in Figs 4 and 5.

## Discussion

As expected, CHF, CTL, and CLF animals exhibited different levels of defensive freezing behavior that was induced by stimuli that were contextually associated with footshock on the previous day. These results indicated that our breeding protocol effectively produced different levels of freezing and confirmed that CHF animals had a higher anxious phenotype and CLF animals had a lower anxious phenotype when both breeding lines were compared to CTL animals [31]. Our results indicate that both CHF and CLF animals exhibited clear differences in Fos immunoreactivity in certain brain structures compared to randomly selected CTL animals.

Fos expression is well stablished as a marker of neuronal activity (for a review, please see [32]), yet some important issues should be considered when interpreting functional mapping

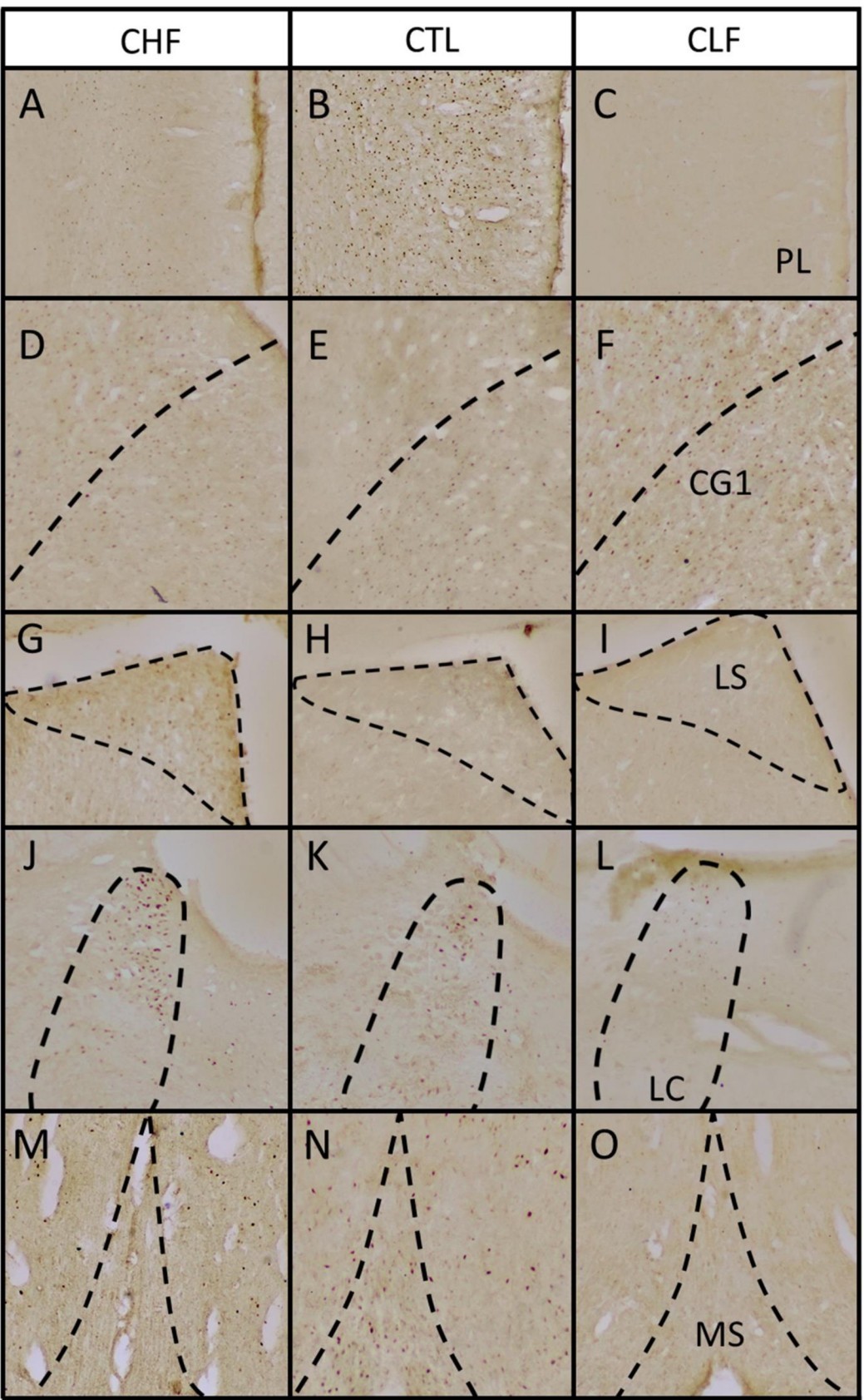

**Fig 4.** (A-O) Representative photomicrographs of coronal sections showing Fos immunoreactivity in brain regions that displayed significant differences in Fos expression between groups in Fig 3. (A, D, G, J, and M) Carioca high freezing (CHF), (B, E, H, K, and N) Carioca low freezing (CLF), and (C, F, I, L, and O) control (CTL) rats. (A-C) PL, prelimbic cortex; (D-F) CG1, anterior cingulate cortex—subregion 1; (G-I) LS, lateral septal nucleus; (J-L) LC, locus coeruleus; (M-O) MS, medial septal nucleus. Dashed lines demarcate the presumptive boundaries of studied brain regions. All images were taken at ×40 magnification.

data based on this technique such as sample size, number of comparisons between groups, stimulus strength, time after stimulus, and potential interference from other intrinsic factors, like physiological noise [33–40]. Another issue worth mentioning relates to the fact that, in the current study, comparisons in Fox expression were not made to Fos-expression levels observed in animals exposed to a new context and/or to naïve animals. Although these would be valuable additions, taking into account the already stated factors that should be weighted when conducting Fos immunoreactivity, we opted to only use, as a control group, animals that were exposed to the same Fos triggering stimulus (i.e. behavioral response to contextual cues previously associated with footshocks). Moreover, along this line of thought, CHF and CLF samples were from rats that had clear and similar levels of high and low freezing responses, respectively (please see ± SE of freezing percentage in Fig 2). In any case, this study is an initial step towards characterizing neural activity changes across multiple brain regions of the Carioca lines in response to contextual fear memories. Future studies are needed to further explore this relationship using Fos immunochemistry, as well as other techniques commonly applied to study neuronal activity.

## Fos expression in CHF animals

An increase in neuronal activation in the PVN was observed in CHF animals when compared to CTL rats. This result is consistent with two other studies that also found an increase in Fos expression in the PVN in another breeding line of rats with high anxiety-like behavior (HAB rats) that were selected based on a low number of open-arm entries in the elevated plus maze [41, 42]. The PVN is the main neural structure that is responsible for regulating hypothalamic-pituitary-adrenal (HPA) axis responses to different types of aversive stimulation [43–45]. Previous studies from our group indicated that CHF animals that were exposed to contextual cues associated with footshock exhibited an increase in circulating serum corticosterone levels compared to CTL and CLF animals [21, 46, 19]. Therefore, our findings indicate that CHF animals exhibit overactivation of neurocircuitries that are responsible for controlling the HPA axis neuroendocrine response to aversive stimuli.

Supporting this result, CHF animals also exhibited an increase in neuronal activity in the LC, the main source of the LC-norepinephrine (LC-NE) system. This system sends wide excitatory projections to different brains structures, including significant connections to PVN [47, 48], indicating a possible excitatory role for the LC-NE system in the HPA axis response [49]. The LC-NE system also plays a central role in autonomic regulation through direct projections to sympathetic preganglionic neurons in the brainstem and spinal cord [47]. These results suggest that CHF animals might also present an imbalance between sympathetic and parasympathetic branches of the autonomic nervous system and the HPA axis. Several studies indicate that generalized anxiety activates both the HPA axis and sympatho-adrenal axis [for review, see 50]. Finally, the LC is also responsible for vigilance arousal activation in response to potentially dangerous stimuli. The disproportional activation of this area may play an important role in the etiology of several anxiety disorders [51].

The septal area is a heterogeneous structure that can be divided into the LS and MS (or diagonal band of Broca nuclei). Each of these areas has behavioral, anatomical, and neurochemical specificities [52–55]. Behavioral and autonomic evidences indicate that both the LS

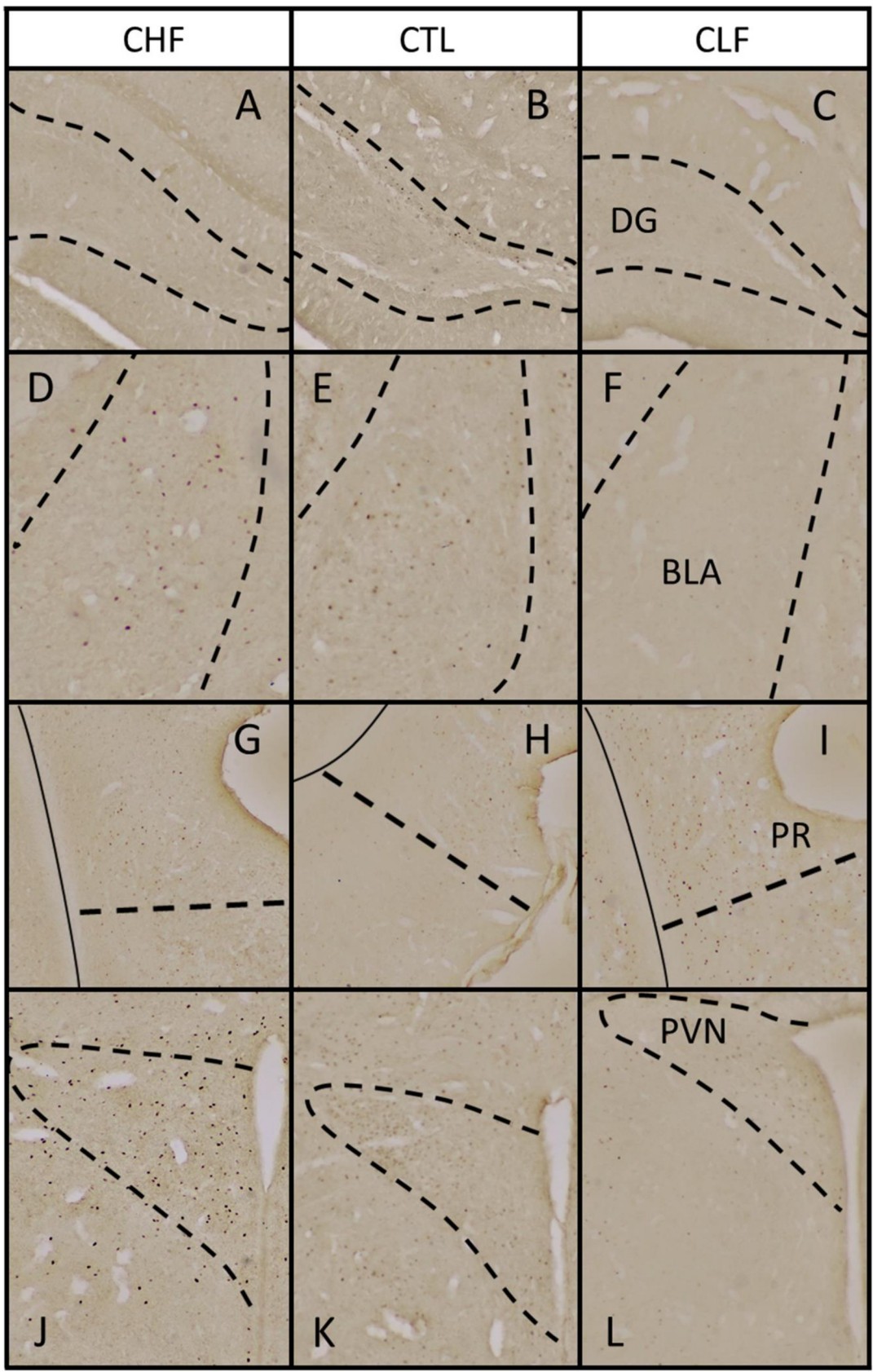

**Fig 5.** (A-L) Representative photomicrographs of coronal sections showing Fos immunoreactivity in brain regions that displayed significant differences in Fos expression between groups in Fig 3. (A, D, G, and J) Carioca high freezing (CHF), (B, E, H, and K) Carioca low freezing (CLF), and (C, F, I, and L) control (CTL) rats. (A-C) DG, dentate gyrus; (D-F) BLA, basolateral amygdaloid nucleus; (G-I) PR, perirhinal cortex; (J-L) PVN, paraventricular hypothalamic nucleus. Dashed lines demarcate the presumptive boundaries of studied brain regions and solid lines in (G-I) indicate white matter. All images were taken at ×40 magnification.

and MS play a critical role in behavioral, autonomic, and neuroendocrine responses to stress [56]. For example, LS lesions decreased and MS lesions increased conditioned freezing in response to contextual cues that were associated with footshock [57]. This behavioral dissociation between the LS and MS in contextual fear conditioning is consistent with the present results. This study found that CHF animals exhibited an increase in Fos expression in the LS and a decrease in Fos expression in the MS. Inhibition of the LS was also shown to reduce contextual fear conditioning, cardiovascular responses and arterial pressure [56] that were induced by reexposure to the aversively conditioned context, suggesting an important excitatory role for the LS in behavioral and autonomic response control in contextual fear conditioning. However, other studies that investigated the role of the LS in neuroendocrine regulation reported findings that were in the opposite direction. Excitotoxic lesions of the LS increased ACTH and corticosterone levels and Fos expression in the PVN in rats that were exposed to the forced swim test [58], suggesting that the LS might play an inhibitory role in the HPA axis response. A possible explanation for this discrepant result might be the fact that different subregions of the LS have opposite influences on the HPA axis. The ventral portion of the LS (vLS) might be responsible for increasing different aspects of fear responses, and the rostral portion of the LS might be engaged in inhibiting defensive responses [59]. Therefore, the increase in Fos expression in the LS in CHF animals might be restricted to the vLS, which has direct excitatory projections to the PVN [60].

Both MS and DG neurons had low Fos expression in CHF animals compared to CTL animals. The MS sends excitatory projections to the DG via the fimbria-fornix [61, 62]. The DG is well known to contribute to glucocorticoid-induced feedback inhibition of the HPA axis to avoid excessive neuroendocrine activity in response to aversive stimuli [43]. For example, animals that were exposed to aversive stimuli exhibited a decrease in glucocorticoid receptor (GR) expression in the DG, which in turn increased neuronal activity in the PVN and HPA axis [63]. Another study selected rats for high (HR) and low (LR) conditioned freezing responses in the contextual fear conditioning paradigm and found that HR rats had lower GR expression levels in the DG [64]. A previous study from our group indicated that CHF animals had fewer newborn neurons (i.e., neuroblasts) in the DG, together with decreases in the number and length of tertiary dendrites [21]. Therefore, an increase in DG neuronal activity in CHF animals might blunt the inhibitory role of the DG in PVN neuron activation, thus producing an exaggerated HPA neuroendocrine response to threatening stimuli.

Surprisingly, CHF animals exhibited a decrease in Fos expression in the PL compared to CTL animals. Several studies indicated that the PL is responsible for the acquisition and expression of conditioned fear [65–69]. However, the PL also appears to play an inhibitory role in the HPA axis. For example, PL stimulation resulted in a decrease in Fos expression in the PVN and reductions of ACTH and corticosterone responses in animals that were exposed to restraint stress [70]. Therefore, the reduction of neuronal activity in the PL in CHF animals in the present study might also be associated with a decrease in the activation of neurocircuitries that are responsible for inhibiting neuroendocrine responses of the HPA axis. The PL does not project directly to the PVN [71]. Therefore, the inhibitory influence of the PL on the HPA axis may be mediated by excitatory projections to the DG.

## Fos expression in CLF animals

CLF animals were selectively bred for low contextual fear conditioning. Therefore, we expected to observe a decrease in the activity of neurocircuitries that are involved in this type of learning. In the present study, CLF animals exhibited hypoactivity of the PVN compared to CTL animals. This result supports the notion that our two bidirectional line of animals have opposite neuroendocrine dysregulation of the HPA axis. However, the neurocircuitry that modulates the activity of PVN neurons appears to be different between CHF and CLF animals. The amygdala plays a pivotal role in neurocircuitry that is involved in contextual fear conditioning [72–74]. Among the different amygdaloid nuclei, the BLA appears to be the main neural substrate of contextual fear conditioning [75, 76]. For example, neurotoxic lesions of the BLA abolished conditional freezing in response to contextual cues that were previously associated with footshock [77–80] and inhibited HPA activity in both acutely and chronically stressed animals [81]. Indeed, evidence indicates that the involvement of neural activation of the BLA during aversive learning depends on HPA activity. Accordingly, an infusion of a glucocorticoid receptor agonist in the BLA after training enhanced the acquisition of aversive memories, whereas a glucocorticoid receptor antagonist impaired the acquisition of aversive memories [82–84]. In the present study, in contrast to CHF animals, CLF animals exhibited a reduction of activation of this amygdaloid nucleus compared to CTL animals. The BLA sends excitatory projections to the PVN [45]. Therefore, the BLA might be less able to modulate the function of the PVN in CLF animals, with a consequent decrease in HPA axis activity during aversive learning.

Low Fos expression was also observed in the PL in CLF animals compared to CTL animals. This result is consistent with the view that CLF animals exhibit a reduction of activation of the neurocircuitry that is responsible for contextual fear conditioning. As discussed above, neurons in the PL play an excitatory role in conditioned fear behavior. Immunochemistry indicated that the PL exhibits greater activation when animals are reexposed to contextual cues that are previously associated with footshock [85]. Moreover, pharmacological inhibition of the PL reduced the expression of conditioned fear [86]. The PL sends descending excitatory projections to the BLA [87], which likely contributed to the decrease in BLA activity in CLF animals.

In the present study, Fos expression increased in the CG1 region of the ACC and the PR in CLF animals compared to CTL animals. An increase in neuronal activity in the CG1 was also found in LAB rats [88]. A study of rats that were selected for high (HR) and low (LR) conditioned freezing responses in the contextual fear conditioning paradigm found that LR rats exhibited an increase in GR expression in the CG1 region [64]. Lesions of the ACC, including the CG1 region, increased plasma levels of both ACTH and corticosterone following restraint stress, indicating an inhibitory role of these neurons in HPA activity [89]. Indeed, there are inhibitory projections from the ACC to the PVN [90]. Thus, the increase in neural activity in the CG1 region in CLF animals might have resulted in intensification of the inhibitory influence of the CG1 on neural activity in the PVN, thus resulting in a decrease in HPA axis activity.

Fos immunochemistry also revealed that CLF animals exhibited higher activity in the PR. The PR is a high-order associative area that combines different sensory modalities to integrate and represent polymodal information. It plays an important role in processing complex stimuli, such as contextual cues [91–93]. The PR and BLA appear to play opposite roles in stimulus processing during contextual fear conditioning [94]. The BLA neurons are activated when the context is dangerous, whereas the PR is activated when the context is safe. This dissociation between the BLA and the PR according to the emotional significance of contextual cues may

be mediated by the CG1. This region also plays a major role in information processing, assigning emotional valence to external stimuli [95, 96]. The CG1 sends direct excitatory projections to the PR [97, 98] and direct inhibitory projections to the BLA [99]. Accordingly, the greater activity of CG1 neurons in CLF animals may contribute to the lower activity of BLA neurons and greater activity of PRC neurons.

Such high neural activity of the PR in CLF animals might indicate an overload of multisensorial information processing, a characteristic that may be involved in attention-deficit/hyperactivity disorder (ADHD). A further support to the proposition that CLF rats could be an animal model of ADHD is the fact that children with a diagnosis with ADHD have an underactivity of the stress system [100]. For example, preschool children with a diagnosis of ADHD were reported to have hypoactivity of the HPA axis [101]. Therefore, the excessive reduction of contextual fear conditioning in CLF animals coupled with the high activity of the PR and low Fos expression in the PVN may represent an animal model of this pathological conditioning. In accordance with this possibility, a research group at the Max Planck Institute of Psychiatry (Munich, Germany) developed a mouse model of extreme trait anxiety, based on selective breeding for low, normal, and high open-arm exploration in the elevated plus maze. Pharmacological and behavioral results indicated that genetically selected low anxiety-related behavior might indeed represent an animal of ADHD [102, 103].

## Conclusions

Several brain structures are involved in behavioral and neurophysiological defense systems that are responsible for adapting an individual to threatening situations. The aim of the present study was to identify significant changes in brain structures in our bidirectional lines of animals that were selected for high (CHF) and low (CLF) freezing in response to contextual cues that were previously associated with footshock. Based on high and low Fos expression in various brain structures in the present study, we propose that our two bidirectional lines exhibit different neural activation patterns that have opposing influences on the PVN. Two pathways might be responsible for high PVN activity in CHF animals. One pathway is associated with an increase in excitatory projections that the PVN receives from the LC and LS. The LC might also activate the sympathetic autonomous system. The other pathway might be associated with a decrease in inhibitory projections that the PVN receives from the DG, which in turn receives projections from the MS and PL. Two other pathways might mediate the decrease in neuronal activity in the PVN in CLF animals. One of the pathways may be associated with a decrease in the activation of structures that are important for contextual fear conditioning, such as the BLA, which sends excitatory projections to the PVN and receives excitatory projections from the PL. The other pathway may be related to the increase in neural activity of the CG1, which sends inhibitory projections to both the BLA and PVN and excitatory projections to the PR. These proposed neurocircuits suggest that both lines might exhibit dysfunctional responses of the HPA axis, the main neuroendocrine system that is involved in the maintenance of homeostasis after exposure to stressful stimuli.

## Supporting information

**S1 Table. Mean (± SEM) number of Fos-positive neurons/0.1 mm$^2$ in control (CTL), high (CHF), and low (CLF) breeding lines of rats in different brain structures after exposure to contextual cues that were previously associated with footshock (7–12 animals per region).** (DOCX)

## Author Contributions

**Conceptualization:** Laura A. León, Marcus L. Brandão, Fernando P. Cardenas, J. Landeira-Fernandez.

**Data curation:** Laura A. León.

**Formal analysis:** Fernando P. Cardenas, Diana Parra, Thomas E. Krahe, Antonio Pedro Mello Cruz, J. Landeira-Fernandez.

**Funding acquisition:** Marcus L. Brandão, J. Landeira-Fernandez.

**Methodology:** Laura A. León, Marcus L. Brandão, Fernando P. Cardenas, Diana Parra, Thomas E. Krahe, Antonio Pedro Mello Cruz, J. Landeira-Fernandez.

**Writing – original draft:** Marcus L. Brandão, Fernando P. Cardenas, Thomas E. Krahe, Antonio Pedro Mello Cruz, J. Landeira-Fernandez.

**Writing – review & editing:** J. Landeira-Fernandez.

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
