## [Decision Letter · Decision Letter 0]

19 Mar 2020

PONE-D-20-05104

Distinct patterns of brain Fos expression in Carioca High- and Low-conditioned Freezing Rats

PLOS ONE

Dear Dr. Landeira-Fernandez,

Thank you for submitting your manuscript to PLOS ONE. After careful consideration, we feel that it has merit but does not fully meet PLOS ONE’s publication criteria as it currently stands. Therefore, we invite you to submit a revised version of the manuscript that addresses the points raised during the review process.

Please especially concentrate on the issue of a control for the two test groups at baseline, prior to FC. 

We would appreciate receiving your revised manuscript by May 03 2020 11:59PM. To enhance the reproducibility of your results, we recommend that if applicable you deposit your laboratory protocols in protocols.io, where a protocol can be assigned its own identifier (DOI) such that it can be cited independently in the future. For instructions see: http://journals.plos.org/plosone/s/submission-guidelines#loc-laboratory-protocols

We look forward to receiving your revised manuscript.

Kind regards,

Alexandra Kavushansky, PhD

Academic Editor

PLOS ONE

Journal Requirements:

2. Thank you for including your ethics statement; "The experimental procedures were performed in accordance with the guidelines for experimental animal research that were established by the Brazilian Society of Neuroscience and Behavior (SBNeC) and National Institutes of Health Guide for the Care and Use of Laboratory Animals. Animal handling and the methods of sacrifice were reviewed and approved by the Committee for Animal Care and Use of PUC-Rio (protocol no. 20/2009)."

Please amend your current ethics statement to confirm that your named ethics committee specifically approved this study.

For additional information about PLOS ONE submissions requirements for ethics oversight of animal work, please refer to http://journals.plos.org/plosone/s/submission-guidelines#loc-animal-research  

4. Your ethics statement must appear in the Methods section of your manuscript. If your ethics statement is written in any section besides the Methods, please move it to the Methods section and delete it from any other section. Please also ensure that your ethics statement is included in your manuscript, as the ethics section of your online submission will not be published alongside your manuscript.

5. Please ensure that you refer to Figure 3 in your text as, if accepted, production will need this reference to link the reader to the figure.

6. Please upload a copy of Figure 5, to which you refer in your text on page 14. If the figure is no longer to be included as part of the submission please remove all reference to it within the text.

7. We note that Figures in your submission might contain copyrighted images. All PLOS content is published under the Creative Commons Attribution License (CC BY 4.0), which means that the manuscript, images, and Supporting Information files will be freely available online, and any third party is permitted to access, download, copy, distribute, and use these materials in any way, even commercially, with proper attribution. For more information, see our copyright guidelines: http://journals.plos.org/plosone/s/licenses-and-copyright.

1.    You may seek permission from the original copyright holder of the Figures to publish the content specifically under the CC BY 4.0 license.

Reviewers' comments:

Reviewer's Responses to Questions

**Comments to the Author**

1. Is the manuscript technically sound, and do the data support the conclusions?

Reviewer #1: Yes

Reviewer #2: Yes

Reviewer #3: Yes

2. Has the statistical analysis been performed appropriately and rigorously? 

Reviewer #1: Yes

Reviewer #2: Yes

Reviewer #3: Yes

3. Have the authors made all data underlying the findings in their manuscript fully available?

Reviewer #1: Yes

Reviewer #2: Yes

Reviewer #3: Yes

4. Is the manuscript presented in an intelligible fashion and written in standard English?

Reviewer #1: Yes

Reviewer #2: Yes

Reviewer #3: Yes

5. Review Comments to the Author

Reviewer #1: In this article, Leon et al examined neuronal activity patterns in selectively bred Carioca High and Low-Freezing rats that exhibited high and low freezing responses, respectively, in response to exposure to a fear-associated context. To that end they performed Fos immunohistochmesitry in a vast array of brain areas that were relevant to fear and anxiety responses including areas such as limbic cortices, hypothalamic and brain stem structures. The authors found differential Fos expression several brain areas between these two rat lines following fear context exposure, such as Fos expression in the opposing direction in the paraventricular nucleus of the hypothalamus (PVN) compared to controls. The authors speculate that in these rat lines, the differential recruitment of limbic brain areas result in opposing PVN activity, which gives rise to differences in conditioned fear responses.

In general, these are interesting findings that shed light on how various limbic brain areas are recruited to control fearful behaviours. The manuscript is clearly written and is easy to follow. Their Fos findings are discussed with regards to the known behavioral roles of the examined brain areas as well as the neuroanatomical connections with the existing literature.

General:

-Please provide page and line numbers. It makes it easier for the reviewer to communicate where certain changes have to be made.

Methods:

-It says ammonia was used to clean the cages, but does it have an aversive quality?

Results:

-Could the authors please provide representative Fos images for some of the key areas that displayed differential expression of Fos, such as the PVN? Also, please indicate using arrows what a typical Fos+ cells looks like.

-Figures 3 and 4:

I believe Fos 'expression' is what is being measured and not 'activity' per se. Please change activity to expression.

Figure legends:

-Please indicate the n number for each experiment.

Discussion:

-The authors compare Fos expression between groups of rats that underwent fear context exposure, and there is no Unpaired control group that received shock exposure in an alternate context, and is exposed to a neutral context on test day. As such, it is difficult to parse out the Fos expression that occurs as a result of fear memory recall versus exposure to a non-home cage context, habituation to the training context, etc. The authors do not need to run any additional experiments, but they could mention this caveat in their discussion and suggest the inclusion of these controls in future experiments.

Reviewer #2: The manuscript by Leon et al reports on the Fos expression pattern in distinct brain areas of Carioca High- and Low-conditioned freezing rats, two breeding lines based on defensive freezing in the contextual fear conditioning paradigm. The hypothalamic paraventricular nucleus appears to be the main region involved in the different pattern of neural activation and inhibition.

The paper is based on the use of an anxiety model began in 2006 and it has been reported by Gomez and Landeira-Fernandez (2008) that after three generations reliable differences between these two lines were already present, indicating a strong heritable component of aversive learning in the contextual fear conditioning paradigm. A weakness of the introduction is the lack of data regarding the construct and predictive validity of this model of anxiety and a comment on this would certainly strenghten the paper.

The experimental procedures are well conducted and results are correctly presented and analyzed. As for the Figures, the authors illustrate the positive cell scores in the different area with a table; representative figures of areas with clear cut differences among lines should be presented in parallel, first of all to illustrate the actual differences among areas, secondly to visibly support the numerical data.

Minor points:

In the Materials and Methods, “primary Fos rabbit polyclonal IgG” should be changed to “rabbit polyclonal primary IgG against Fos”.

Reviewer #3: The study attempts to characterize neural circuits that are different between high and low freezing rats. The authors identify a number of substrates that are different between CHL and control and CFL and control rats. The manuscript is well written, straightforward, and easy to understand what the authors are trying to communicate.

6. PLOS authors have the option to publish the peer review history of their article (what does this mean?). If published, this will include your full peer review and any attached files.

Reviewer #1: No

Reviewer #2: No

Reviewer #3: No

---

## [Author Response · Author response to Decision Letter 0]

14 May 2020

RESPONSE TO THE REVIEWERS

Please find a revised manuscript entitled “Distinct patterns of brain Fos expression in Carioca High- and Low-conditioned Freezing Rats" (Manuscript ID: PONE-D-20-05104). We would like to thank the reviewers for their thorough reading and insightful comments. We include a point-by-point response to the reviewers’ concerns. For your convenience in our response we also provide the amended text as it appears in the revised manuscript.

REVIEWER #1:

We were extremely glad to see that the Reviewer considered that our work “(…) In general, these are interesting findings that shed light on how various limbic brain areas are recruited to control fearful behaviours. The manuscript is clearly written and is easy to follow.” The Reviewer also presented some concerns that need to be addressed. Here, we considered more fully all his/her comments.

General:

Please provide page and line numbers. It makes it easier for the reviewer to communicate where certain changes have to be made.

We agree. Page and line numbers were added to the revised version of the manuscript.

Methods:

It says ammonia was used to clean the cages, but does it have an aversive quality?

The referee raises an interesting concern. The concentration of ammonia used to clean the cages was small (5 %) and has been used before by our group and others (Hassan et al., 2003; Macêdo-Souza et al., 2020; Sanders & Fanselow, 2003; Young et al., 1994). Further, we waited for the chamber to be complete dry before testing another animal.

Hassan W, Gomes Vde C, Pinton S, Batista Teixeira da Rocha J, Landeira-Fernandez J. Association between oxidative stress and contextual fear conditioning in Carioca high- and low-conditioned freezing rats. Brain Res., 1512:60-7, 2013.

Macêdo-Souza C, Maisonnette SS, Filgueiras CC, Landeira-Fernandez J, Krahe TE. Cued Fear Conditioning in Carioca High- and Low-Conditioned Freezing Rats. Front Behav Neurosci. 24;13:285, 2020.

Sanders MJ, Fanselow MS. Pre-training prevents context fear conditioning deficits produced by hippocampal NMDA receptor blockade. Neurobiol Learn Mem. 80(2):123-9, 2003.

Young SL, Bohenek DL, Fanselow MS. NMDA processes mediate anterograde amnesia of contextual fear conditioning induced by hippocampal damage: immunization against amnesia by context preexposure. Behav Neurosci., 108(1):19-29, 1994.

Results:

Could the authors please provide representative Fos images for some of the key areas that displayed differential expression of Fos, such as the PVN? Also, please indicate using arrows what a typical Fos+ cells looks like.

The reviewer raises an important concern. We now provide representative photomicrographs of c-Fos expression in different brain regions for each experimental group. Revised version, Figures 4 and 5.

Figures 3 and 4:

I believe Fos 'expression' is what is being measured and not 'activity' per se. Please change activity to expression.

The reviewer is correct. However, Figures 3 and 4 were removed from the revised version of the manuscript as per suggestion from another reviewer.

Figure legends:

Please indicate the n number for each experiment.

We thank the reviewer for bringing this matter to our attention. This information has been added to the main text and to the figure legends of the revised manuscript (Methods, pages 4 and 5, lines 118-121; Methods, page 6, last paragraph, line 184).

“We used male adult rats selectively bred for high (CHF, n = 10) and low (CLF, n = 10) contextual fear conditioning according to previously described procedures (Castro-Gomes and Landeira-Fernandez, 2008). Non-selectively bred Wistar rats were used a control group (CTL, n = 12).”

“The system was calibrated to ignore background staining. All brain regions were counted bilaterally (7-12 animals per region), and the mean was calculated for each structure.”

“Figure 2. Mean ± SEM percentage of time spent freezing in Carioca high freezing (CHF, n = 10), Carioca low freezing (CLF, n = 10), and control (CTL, n = 12) rats. Graph depicts freezing responses 24 h after exposure to contextual cues that were previously associated with footshocks.”

“Figure 3. (A-B) Fos expression in different brain structures of Carioca high freezing (CHF), Carioca low freezing (CLF), and control (CTL) animals after exposure to contextual cues that were previously associated with footshocks (7-12 animals per region per group)…”

“Supplemental Table 1 - Mean (± SEM) number of Fos-positive neurons/0.1 mm2 in control (CTL), high (CHF), and low (CLF) breeding lines of rats in different brain structures after exposure to contextual cues that were previously associated with footshock (7-12 animals per region).”

Discussion:

The authors compare Fos expression between groups of rats that underwent fear context exposure, and there is no Unpaired control group that received shock exposure in an alternate context, and is exposed to a neutral context on test day. As such, it is difficult to parse out the Fos expression that occurs as a result of fear memory recall versus exposure to a non-home cage context, habituation to the training context, etc. The authors do not need to run any additional experiments, but they could mention this caveat in their discussion and suggest the inclusion of these controls in future experiments.

We thank the reviewer for bringing up this important issue to our attention. We now address this issue in the revised version of the discussion (Discussion, pages 9, first paragraph, lines 254-272).

It is also worth to mention that due to the COVID-19 pandemic, non-essential research activities were put on hold and our animal facility is operating in a limited capacity with focus on the health and welfare of the Carioca lines. Furthermore, new experiments would require a substantial number of animals, not only those accounted for new experimental groups, but also for representative samples of each original group. We believe that the addition of such number of animals to the study is not in line with NIH guidelines of animal care (https://www.ncbi.nlm.nih.gov/books/NBK54045/). 

“Fos expression is well stablished as a marker of neuronal activity (for a review, please see Dampney and Horiuchi, 2003), yet some important issues should be considered when interpreting functional mapping data based on this technique such as sample size, number of comparisons between groups, stimulus strength, time after stimulus, and potential interference from other intrinsic factors, like physiological noise (Bullitt et al., 1992; Chamberlin and Saper, 1994; French et al., 2001; Guzowski et al., 2006; He et al., 2018; Lima and Avelino, 1994; Nakamura and Morrison, 2010; Saleh and Connell, 1998). Another issue worth mentioning relates to the fact that, in the current study, comparisons in Fox expression were not made to Fos-expression levels observed in animals exposed to a new context and/or to naïve animals. Although these would be valuable additions, taking into account the already stated factors that should be weighted when conducting Fos immunoreactivity, we opted to only use, as a control group, animals that were exposed to the same Fos triggering stimulus (i.e. behavioral response to contextual cues previously associated with footshocks). Moreover, along this line of thought, CHF and CLF samples were from rats that had clear and similar levels of high and low freezing responses, respectively (please see ± SE of freezing percentage in Figure 2). In any case, this study is an initial step towards characterizing neural activity changes across multiple brain regions of the Carioca lines in response to contextual fear memories. Future studies are needed to further explore this relationship using Fos immunochemistry, as well as other techniques commonly applied to study neuronal activity.”

REVIEWER #2

We were thankful for his/her insightful comments and for the fact that “(…) The experimental procedures are well conducted and results are correctly presented and analyzed.” However, the reviewer also presented some concerns that need to be addressed.

Major:

A weakness of the introduction is the lack of data regarding the construct and predictive validity of this model of anxiety and a comment on this would certainly strengthen the paper.

We agree. Our paper lacked a characterization of the Carioca lines regarding its construct and predictive validity as a model of anxiety. The introduction has been revised to include this information (Introduction, pages 3 and 4, lines 79-102).

“Over the last decade, several works have established the Carioca lines as an animal model for the study of anxiety related disorders. For instance, CHF rats display more anxious like behaviors than CLF rats in the elevated plus maze, less social interactions than normal animals, and higher plasma corticosterone concentrations compared to both CLF and CTL animals (Dias et al., 2009; Mousovich-Neto et al., 2015; Salviano et al., 2014). On the other hand, no behavioral differences were found between CHF and CLF animals in the forced swim test (Dias et al., 2009), suggesting a dissociation between anxiety and depression traits in the Carioca lines. Moreover, cognitive and memory performance of CHF and CLF rats both in the object recognition task and the Morris water maze test were similar to normal animals (Dias et al., 2009; Dias et al., 2014).”

“The freezing response to contextual cues previously associated with footshocks observed in the Carioca lines has also been pharmacologically validated as an adequate model of anxiety disorders. We have recently demonstrated that classic anxiolytic benzodiazepines, such as midazolam reduces the amount of conditioned contextual freezing responses of CHF rats (Cavaliere et al., 2020). Similar results were observed with non-benzodiazepine anxiolytics, microinjections of ketaserin, a 5-HT2A/2C receptor antagonist, in the infralimbic mPFC cortex reduced the amount of freezing responses of CHF rats in the contextual fear conditioning paradigm (León et al., 2017). Moreover, both systemic and infralimbic cortical injections of ketanserin increased the number of open arm entries and time spent in the open arms of CHF animals in the elevated plus maze (León et al., 2017).”

“According to these behavioral and pharmacological findings, it is likely that CHF and CLF rats display different neural activity patterns in response to contextual cues previously associated with footshocks. One way to explore this possibility is to evaluate changes in Fos expression in specific brain regions.”

As for the Figures, the authors illustrate the positive cell scores in the different area with a table; representative figures of areas with clear cut differences among lines should be presented in parallel, first of all to illustrate the actual differences among areas, secondly to visibly support the numerical data.

We agree with the reviewer. We now provide representative photomicrographs of c-Fos expression in different brain regions for each experimental group. Revised version, Figures 4 and 5.

Minor:

In the Materials and Methods, “primary Fos rabbit polyclonal IgG” should be changed to “rabbit polyclonal primary IgG against Fos”.

The reviewer is correct. We have modified the manuscript accordingly (Methods, page 6, first paragraph, lines 169-170).

RIVEWER #3

We were pleased that this reviewer indicated that “(…) The manuscript is well written, straightforward, and easy to understand what the authors are trying to communicate.” However, he/she also indicated that some issues in still need clarification. We have addressed in the revised manuscript each of these issues, as described below:

Major concerns:

1. The strain of rat that is used for CHL and CFL is never mentioned.

The reviewer raises an important concern. We now state in the revised version of the manuscript that CHF, CLF and control animals are from the Wistar strain (Methods, pages 4 and 5, lines 118-121).

“We used male adult rats selectively bred for high (CHF, n = 10) and low (CLF, n = 10) contextual fear conditioning according to previously described procedures (Castro-Gomes and Landeira-Fernandez, 2008). Non-selectively bred Wistar rats were used as a control group (CTL, n = 12).”

2. In the abstract and Introduction the authors need to make it clear that they are examining c-Fos expression in response to contextual fear conditioning. To this end the authors should have a group of rats that were never conditioned or represent baseline differences. How sure are the authors that levels of c-Fos observed do not represent basal differences between the groups of animals?

The referee raises an important concern. While we agree that the inclusion of the suggested control groups could improve and strengthen the conclusions of our work, based on the reasons listed below we respectfully decided against carrying out new experiments.

Different from knockout and knock-in animals, every generation of both Carioca lines exhibit a wider degree of behavioral-phenotype variability (i.e. CHF animals that do not show high levels of freezing and CLF ones that freeze). Thus, and acknowledging that many factors contribute to data variability regarding expression of early genes (Dampney and Horiuchi, 2003), we opted to use only CHF and CLF animals with respectively high and low levels of freezing behavior (please see ± SE of freezing percentage in Figure 2). By doing so, we believe we are minimizing within group variability. Unfortunately, the same is not true when using naïve animals. Add to this, the fact that Fos expression is known to be influenced by factors such as stimulus strength, time, and physiological functions (French et al., 2001; Guzowski et al., 2006; He et al., 2018; Bullitt et al., 1992; Lima and Avelino, 1994; Chamberlin and Saper, 1994; Nakamura and Morrison, 2010; Saleh and Connell, 1998). Therefore, one can predict that naive animals would present a greater within group variability compared to CHF, CLF, and CTL animals selected by their freezing response to contextual cues previously associated to footshocks. One measure to minimize this issue would be to increase the number of animals in naïve experimental groups. However, as mentioned below, we have some restrictions concerning this approach. Another option, would be to use, as controls, animals that were randomly bred irrespective of their freezing behavior and were exposed to the same Fos triggering stimulus (i.e. behavioral response to contextual cues previously associated with footshocks – CTL group.

Last but not least, our research activities were put on hold due to the COVID-19 pandemic. Our animal facility is operating in a limited capacity with focus on the health and welfare of the Carioca lines. Furthermore, conducting extra experiments would require, not only animals to perform the suggested experiments, but also representative samples of animals for the original experimental groups. In our estimates, that would account for as many as or more animals than we have already used, which one might argue that is not in consonance with the NIH guidelines of animal care (https://www.ncbi.nlm.nih.gov/books/NBK54045/).

Nonetheless, regardless limitations and issues, we agree with the reviewer that the inclusion of naive animals would have been of great value. As suggested by another reviewer, we now stress out some of the aforementioned points in the discussion of the revised manuscript (Discussion, pages 9, first paragraph, lines 254-272).

“Fos expression is well stablished as a marker of neuronal activity (for a review, please see Dampney and Horiuchi, 2003), yet some important issues should be considered when interpreting functional mapping data based on this technique such as sample size, number of comparisons between groups, stimulus strength, time after stimulus, and potential interference from other intrinsic factors, like physiological noise (Bullitt et al., 1992; Chamberlin and Saper, 1994; French et al., 2001; Guzowski et al., 2006; He et al., 2018; Lima and Avelino, 1994; Nakamura and Morrison, 2010; Saleh and Connell, 1998). Another issue worth mentioning relates to the fact that, in the current study, comparisons in Fox expression were not made to Fos-expression levels observed in animals exposed to a new context and/or to naïve animals. Although these would be valuable additions, taking into account the already stated factors that should be weighted when conducting Fos immunoreactivity, we opted to only use, as a control group, animals that were exposed to the same Fos triggering stimulus (i.e. behavioral response to contextual cues previously associated with footshocks). Moreover, along this line of thought, CHF and CLF samples were from rats that had clear and similar levels of high and low freezing responses, respectively (please see ± SE of freezing percentage in Figure 2). In any case, this study is an initial step towards characterizing neural activity changes across multiple brain regions of the Carioca lines in response to contextual fear memories. Future studies are needed to further explore this relationship using Fos immunochemistry, as well as other techniques commonly applied to study neuronal activity.”

Bullitt E, Lee CL, Light AR, Willcockson H. The effect of stimulus duration on noxious-stimulus induced c-fos expression in the rodent spinal cord. Brain Res. 1992 May 15;580(1-2):172-9.

Chamberlin NL, Saper CB. Topographic organization of respiratory responses to glutamate microstimulation of the parabrachial nucleus in the rat. J Neurosci 1994;14:6500–10.

Dampney, RAL and Horiuchi J. Functional organization of central cardiovascular pathways: studies using c-fos gene expression. Progress in Neurobiology 71 (2003) 359–384.

French P, O'Connor V, Jones M, Davis S, Errington M, Voss K, et al. Subfield‐specific immediate early gene expression associated with hippocampal long‐term potentiation in vivo. Eur J Neurosci. 2001; 13: 968-976.

Guzowski JF, Miyashita T, Chawla MK, Sanderson J, Maes LI, Houston FP, et al. Recent behavioral history modifies coupling between cell activity and Arc gene transcription in hippocampal CA1 neurons. Proc Natl Acad Sci U S A. 2006; 103: 1077-1082.

He Q, Wang J, Hu H. Illuminating the activated brain: Emerging activity-dependent tools to capture and control functional neural circuits. Neurosci Bull. 2018: 1-9.

Lima D, Avelino A. Spinal c-fos expression is differentially induced by brief or persistent noxious stimulation. Neuroreport. 1994 Oct 3;5(15):1853-6.

Nakamura K, Morrison SF. A thermosensory pathway mediating heat-defense responses. Proc Natl Acad Sci U S A 2010;107:8848–53. 

Saleh TM, Connell BJ. The parabrachial nucleus mediates the decreased cardiac baroreflex sensitivity observed following short-term visceral afferent activation. Neuroscience 1998;87:135–46.

3. An image of the c-Fos immunohistochemistry needs to be provided

The reviewer is correct. We now provide representative photomicrographs of c-Fos expression in different brain regions for each experimental group. Revised version, Figures 4 and 5.

4. The authors do not have enough information to make a model about connectivity between brain regions (e.g. Figures 3 and 4). These figures should be excluded.

We agree with the referee and Figures 3 and 4 from the original version of the manuscript were removed.

5. The information presented in the table should be presented in a graph format. This makes it easier to assess the findings.

We thank the reviewer for bringing this matter to our attention. We now present the results depicted in Table 1 in a graph format (new Figure 3). Data from Table 1 sill available as supplemental material.

6. Discussion is too long and needs to be tempered. The nature of the results (e.g. no difference in Fos in amygdala brain regions between groups of animals) does not allow for strong statements about the role of these brain regions in trait anxiety.

We thank the reviewer for bringing this matter to our attention. The discussion of the revised version has been tempered in line with the referee suggestion. Moreover, we believe that the removal of original Figures 3 and 4, as per suggestion of this same reviewer, helped to tone down the discussion of the revised version of the manuscript.

The following section was removed:

“Importantly, other brain structures that are involved in contextual fear conditioning, such as the PAG and the amygdaloid complex (Canteras et al., 2010; Fanselow, 1994; Kim et al., 2013; Kim and Jung, 2006; Misslin, 2003), did not present any differences in CHF animals compared to CTL or CLF animals. These findings are in agreement with previous results from our group indicating that amygdaloid lesions proportionally reduced the level of contextual fear conditioning in both CHF and CTL animals, thus strengthening the idea that this structure, that is related to fear conditioning, does not play an important role in high trait anxiety-like behavior in CHF animals (Castro-Gomes and Landeira-Fernandez, 2008). Moreover, these results converge with the findings of other studies that employed two breeding lines of rats that were selected based on high (LAB) and low (HAB) open-arm entries in the elevated plus maze (Kalisch et al., 2004; Salomé et al., 2004). HAB and LAB animals did not present any differences in Fos expression in the amygdala or PAG when the animals were exposed to different aversive situations (Salomé et al., 2004).”

7. The PL is critical for expression of fear, but most people would not propose that the PL is critical for acquisition of fear memory. Not sure if the citations the authors are providing for this in the Discussion supports this statement.

We apologized for the mistake. We now provide new references supporting our statements. The following references were included in the revised version of the manuscript:

Gilmartin MR, Kwapis JL, Helmstetter FJ. Trace and contextual fear conditioning are impaired following unilateral microinjection of muscimol in the ventral hippocampus or amygdala, but not the medial prefrontal cortex. Neurobiol Learn Mem. 2012 May;97(4):452-64.

Han CJ, O'Tuathaigh CM, van Trigt L, Quinn JJ, Fanselow MS, Mongeau R, Koch C, Anderson DJ. Trace but not delay fear conditioning requires attention and the anterior cingulate cortex. Proc Natl Acad Sci U S A. 2003 Oct 28;100(22):13087-92.

Robinson-Drummer PA, Heroux NA, Stanton ME. Antagonism of muscarinic acetylcholine receptors in medial prefrontal cortex disrupts the context preexposure facilitation effect. Neurobiol Learn Mem. 2017 Sep;143:27-35.

Twining RC, Lepak K, Kirry AJ, Gilmartin MR. Ventral Hippocampal Input to the Prelimbic Cortex Dissociates the Context from the Cue Association in Trace Fear Memory. J Neurosci. 2020 Apr 15;40(16):3217-3230.

Yang ST, Shi Y, Wang Q, Peng JY, Li BM. Neuronal representation of working memory in the medial prefrontal cortex of rats. Mol Brain. 2014 Aug 28;7:61.

And the following references were removed:

Giustino TF, Maren S (2015) The role of the medial prefrontal cortex in the conditioning and extinction of fear. Front Behav Neurosci 9:298.

Maren S, Phan KL, Liberzon I (2013) The contextual brain: implications for fear conditioning, extinction and psychopathology. Nat Rev Neurosci 14:417-428.

Sotres-Bayon F, Quirk GJ (2010) Prefrontal control of fear: more than just extinction. Curr Opin Neurobiol 20:231-235.

Minor concerns:

1. Some references from the Blanchards, Lang, and Davis are missing.

We thank the reviewer for bringing issue to our attention. The following references were included in the revised version of the manuscript:

Blanchard DC, Griebel G, Pobbe R, Blanchard RJ. (2011). Risk assessment as an evolved threat detection and analysis process. Neurosci Biobehav Rev. 35:991-998.

Blanchard DC, Hynd AL, Minke KA, Minemoto T, Blanchard RJ (2001) Human defensive behaviors to threat scenarios show parallels to fear- and anxiety-related defense patterns of non-human mammals. Neurosci Biobehav Rev. 25(7-8):761-70.

Lang PJ, Davis M, Ohman A. (2000). Fear and anxiety: animal models and human cognitive psychophysiology J Affect Disord. 61: 137-59.

2. Better introduction as to why contextual fear conditioning is a good model of anxiety is needed.

The reviewer raises an important point. This information was included in the introduction of the revised version of the manuscript together with the construct and predictive validity of the Carioca lines as a model of anxiety (Introduction, pages 3, second paragraph, lines 64-74; Introduction, pages 3 and 4, lines 79-102).

“Several rodent models have been developed to investigate the possible etiological mechanisms that underlie anxiety disorders (Haller and Alicki, 2012; Steimer, 2011). Among these, fear conditioning in response to contextual cues is a well-studied experimental model of the aversive expectations of danger that are observed in patients suffering from generalized anxiety disorder (Blanchard et al., 2001; Galvão et al., 2011). Furthermore, considerable evidence indicates that contextual fear conditioning in rats involves neural circuitries similar to that associated with anxiety disorders in humans (Indovina et al., 2011; Kim and Fanselow, 1992; LeDoux, 2000). In this model, rodents receive brief footshocks (unconditioned stimuli) minutes after being placed in a novel chamber, and, when returned to the same chamber 24 h later, they present a typical freezing response to contextual cues associated with the footshocks (González et al., 2003; Landeira-Fernandez, 1996).”

“Over the last decade, several works have established the Carioca lines as an animal model for the study of anxiety related disorders. For instance, CHF rats display more anxious like behaviors than CLF rats in the elevated plus maze, less social interactions than normal animals, and higher plasma corticosterone concentrations compared to both CLF and CTL animals (Dias et al., 2009; Mousovich-Neto et al., 2015; Salviano et al., 2014). On the other hand, no behavioral differences were found between CHF and CLF animals in the forced swim test (Dias et al., 2009), suggesting a dissociation between anxiety and depression traits in the Carioca lines. Moreover, cognitive and memory performance of CHF and CLF rats both in the object recognition task and the Morris water maze test were similar to normal animals (Dias et al., 2009; Dias et al., 2014).”

“The freezing response to contextual cues previously associated with footshocks observed in the Carioca lines has also been pharmacologically validated as an adequate model of anxiety disorders. We have recently demonstrated that classic anxiolytic benzodiazepines, such as midazolam reduces the amount of conditioned contextual freezing responses of CHF rats (Cavaliere et al., 2020). Similar results were observed with non-benzodiazepine anxiolytics, microinjections of ketaserin, a 5-HT2A/2C receptor antagonist, in the infralimbic mPFC cortex reduced the amount of freezing responses of CHF rats in the contextual fear conditioning paradigm (León et al., 2017). Moreover, both systemic and infralimbic cortical injections of ketanserin increased the number of open arm entries and time spent in the open arms of CHF animals in the elevated plus maze (León et al., 2017).”

“According to these behavioral and pharmacological findings, it is likely that CHF and CLF rats display different neural activity patterns in response to contextual cues previously associated with footshocks. One way to explore this possibility is to evaluate changes in Fos expression in specific brain regions.”

---

## [Decision Letter · Decision Letter 1]

3 Jun 2020

PONE-D-20-05104R1

Distinct patterns of brain Fos expression in Carioca High- and Low-conditioned Freezing Rats

PLOS ONE

Dear Dr. Landeira-Fernandez,

Thank you for submitting your manuscript to PLOS ONE. After careful consideration, we feel that it has merit but does not fully meet PLOS ONE’s publication criteria as it currently stands. Therefore, we invite you to submit a revised version of the manuscript that addresses the points raised during the review process.

Please address the concern of the reviewer, considering the potential problem of background illumination in obtaining the the photos of immunostaining.  

We look forward to receiving your revised manuscript.

Kind regards,

Alexandra Kavushansky, PhD

Academic Editor

PLOS ONE

Reviewers' comments:

Reviewer's Responses to Questions

**Comments to the Author**

1. If the authors have adequately addressed your comments raised in a previous round of review and you feel that this manuscript is now acceptable for publication, you may indicate that here to bypass the “Comments to the Author” section, enter your conflict of interest statement in the “Confidential to Editor” section, and submit your "Accept" recommendation.

Reviewer #1: All comments have been addressed

Reviewer #2: All comments have been addressed

2. Is the manuscript technically sound, and do the data support the conclusions?

Reviewer #1: Yes

Reviewer #2: Yes

3. Has the statistical analysis been performed appropriately and rigorously? 

Reviewer #1: Yes

Reviewer #2: Yes

4. Have the authors made all data underlying the findings in their manuscript fully available?

Reviewer #1: Yes

Reviewer #2: Yes

5. Is the manuscript presented in an intelligible fashion and written in standard English?

Reviewer #1: Yes

Reviewer #2: Yes

6. Review Comments to the Author

Reviewer #1: The authors have addressed the reviewer comments adequately and the manuscript has greatly improved. I am satisfied with the revisions.

Reviewer #2: The paper has been greatly improved. The added photomicrographs fill the necessary morphological gap to document the results. The c-fos immunoreaction is fine and nuclei are clearly distinguishable, however I have a concern regarding the background illumination that greatly penalize the immunostaining; some of the photos are gloomy with inhomogenous illumination. Authors should consider to replate the photographs once they have an adequate illumination.

7. PLOS authors have the option to publish the peer review history of their article (what does this mean?). If published, this will include your full peer review and any attached files.

Reviewer #1: No

Reviewer #2: No

---

## [Author Response · Author response to Decision Letter 1]

17 Jun 2020

Please find a revised manuscript entitled “Distinct patterns of brain Fos expression in Carioca High- and Low-conditioned Freezing Rats" (Manuscript ID: PONE-D-20-05104R2). Once again, we would like to thank the reviewer for calling our attention to the immunostaining photographs.

RIVEWER #2

The paper has been greatly improved. The added photomicrographs fill the necessary morphological gap to document the results. The c-fos immunoreaction is fine and nuclei are clearly distinguishable, however I have a concern regarding the background illumination that greatly penalize the immunostaining; some of the photos are gloomy with inhomogenous illumination. Authors should consider to replate the photographs once they have an adequate illumination.

Response: Done

---

## [Decision Letter · Decision Letter 2]

29 Jun 2020

Distinct patterns of brain Fos expression in Carioca High- and Low-conditioned Freezing Rats

PONE-D-20-05104R2

Dear Dr. Landeira-Fernandez,

We’re pleased to inform you that your manuscript has been judged scientifically suitable for publication and will be formally accepted for publication once it meets all outstanding technical requirements.

Kind regards,

Alexandra Kavushansky, PhD

Academic Editor

PLOS ONE

Additional Editor Comments (optional):

Reviewers' comments:

Reviewer's Responses to Questions

**Comments to the Author**

1. If the authors have adequately addressed your comments raised in a previous round of review and you feel that this manuscript is now acceptable for publication, you may indicate that here to bypass the “Comments to the Author” section, enter your conflict of interest statement in the “Confidential to Editor” section, and submit your "Accept" recommendation.

Reviewer #2: All comments have been addressed

2. Is the manuscript technically sound, and do the data support the conclusions?

Reviewer #2: Yes

3. Has the statistical analysis been performed appropriately and rigorously? 

Reviewer #2: Yes

4. Have the authors made all data underlying the findings in their manuscript fully available?

Reviewer #2: Yes

5. Is the manuscript presented in an intelligible fashion and written in standard English?

Reviewer #2: Yes

6. Review Comments to the Author

Reviewer #2: The photoograph backgrounds have been corrected. The manuscript has been greatly improved and now can be accepted as it is.

7. PLOS authors have the option to publish the peer review history of their article (what does this mean?). If published, this will include your full peer review and any attached files.

Reviewer #2: No

---

## [Editor Report · Acceptance letter]

6 Jul 2020

PONE-D-20-05104R2 

Distinct patterns of brain Fos expression in Carioca High- and Low-conditioned Freezing Rats 

Dear Dr. Landeira-Fernandez:

I'm pleased to inform you that your manuscript has been deemed suitable for publication in PLOS ONE. Congratulations! Your manuscript is now with our production department. 

Kind regards, 

on behalf of

Dr. Alexandra Kavushansky 

Academic Editor

PLOS ONE